# Category-based and Popularity-guided Video Game Recommendation: A Balance-oriented Framework

## ABSTRACT

In recent years, the video game industry has experienced substantial growth, presenting players with a vast array of game choices. This surge in options has spurred the need for a specialized recommender system tailored for video games. However, current video game recommendation approaches tend to prioritize accuracy over diversity, potentially leading to unvaried game suggestions. In addition, the existing game recommendation methods commonly lack the ability to establish strict connections between games to enhance accuracy. Furthermore, many existing diversity-focused methods fail to leverage crucial item information, such as item category and popularity during neighbor modeling and message propagation. To address these challenges, we introduce a novel framework, called CPGRec, comprising three modules, namely accuracy-driven, diversity-driven, and comprehensive modules. The first module extends the state-of-the-art accuracy-focused game recommendation method by connecting games in a more stringent manner to enhance recommendation accuracy. The second module connects neighbors with diverse categories within the proposed game graph and harnesses the advantages of popular game nodes to amplify the influence of long-tail games within the player-game bipartite graph, thereby enriching recommendation diversity. The third module combines the above two modules and employs a new negative-sample rating score reweighting method to balance accuracy and diversity. Experimental results on the Steam dataset demonstrate the effectiveness of our proposed method in improving game recommendations. The dataset and source codes are anonymously released at: https://github.com/CPGRec2024/CPGRec.git.

## CCS CONCEPTS

• **Information systems** → **Recommender systems**; *Information extraction*; Personalization;

## KEYWORDS

video game recommendation, accuracy and diversity, item category and popularity, long-tail games

**ACM Reference Format:**
Anonymous submission. 2018. Category-based and Popularity-guided Video Game Recommendation: A Balance-oriented Framework. In *Proceedings of Make sure to enter the correct conference title from your rights confirmation emai (Conference acronym 'XX).* ACM, New York, NY, USA, 10 pages. https://doi.org/XXXXXXX.XXXXXXX

## 1 INTRODUCTION

Recommender systems, essential for personalizing content in digital media, have seen widespread adoption in various domains like e-commerce [8, 9, 15–17, 28], news [21, 29, 30, 33, 39, 46], and music [10, 20, 23, 26, 35, 43]. In recent years, there has been a remarkable surge in the growth of the gaming industry. For instance, an astounding 12,939 games were launched on Steam in 2022, a staggering 43-fold increase compared to the figures from 2012. The large variety of game items underscores the immense potential and importance of crafting recommender systems tailored specifically for the gaming domain. These systems can improve user gaming experiences and boost profits for platforms, developers, and publishers, creating a mutually beneficial cycle.

Recently, the field of game recommendation research has witnessed a surge in scholarly interest, with a predominant emphasis on developing accuracy-focused methods designed to deliver highly personalized game recommendations [3, 4, 6, 14, 25, 40]. While these methods have indeed made significant strides in enhancing accuracy, they often overlook the crucial aspect of recommendation list diversity. This oversight raises concerns about the echo chamber/filter bubble effect [12, 19, 22, 38], where users are confined to familiar games. Such confinement limits their exposure to diverse perspectives and has the potential to reinforce biases. The integration of diversity into recommendations can enrich user experiences by introducing fresh and unexpected choices, broadening their horizons, and simultaneously catering to a more extensive user base. Building upon these insights, we can identify three key challenges currently confronting the field of game recommendation research.

- Traditional accuracy-oriented game recommendation methods [2, 3, 6, 25] primarily address data preprocessing tasks and do not effectively harness the available category information of games. It is essential to emphasize that games exhibit distinctions not only in genres but also in terms of their publishers or developers. Such varying categories can exert significant influence over users' game selection preferences. The State-Of-The-Art (SOTA) game recommendation method, SCGRec [40], explores these diverse game categories, leveraging raw connections within the game graph based on a single category to enhance recommendation accuracy. However, games sharing the same genre may stem from different publishers and developers, implying that relying solely on a single category as a basis for connection may not necessarily exhibit high inherent similarity among games, potentially resulting in accuracy degradation.
- In recent diversity-oriented recommender system research, Graph Neural Network (GNN)-based approaches have emerged as promising solutions. Notably, techniques like dynamic neighbor sampling [42, 47], neighbor selection [41], and adaptations to the Bayesian Personalized Ranking (BPR) Loss calculation method [27] have advanced diversity. However, the

existing methods exhibit two key limitations: (1) the risk of complexity and underutilization of item information in the neighbor modeling techniques, and (2) the long-tail challenge in the message propagation process. In terms of the first limitation, the current methods, which rely on dynamic neighbor sampling or neighbor selection, can become excessively complex with frequent neighbor updates and may struggle to ensure diversity with infrequent updates. Some neighbor selection approaches may overlook vital item attributes like category, as they primarily rely on similarity metrics derived from embeddings, potentially missing important categorical distinctions. In terms of the second limitation, most existing diversity-focused methods operate independently of item popularity information. Consequently, they may lean towards only highlighting popular items during the message propagation process and neglect less popular items within the long-tail distribution, as depicted in Figure 1, impacting the comprehensiveness of recommendations and missing out on the value of unique or less mainstream options.

- The accuracy-diversity dilemma poses a challenge in recommendation systems, demanding a delicate balance between optimizing accuracy and diversity. Striking this balance can be difficult since these aspects often display an inverse relationship. In the current recommendation systems, the BPR loss function is widely employed in accuracy-driven methods [32, 36, 40]. BPR loss utilizes the negative samples to enhance accuracy by guiding modeling from the negative perspective. Conversely, diversity-driven methods typically incorporate loss reweighting or sample probability reweighting into their loss functions [5, 41, 45, 47]. It is worth noting that these approaches predominantly prioritize either accuracy or diversity, struggling to effectively balance both.

To effectively address the triple challenges concerning accuracy, diversity, and the combined accuracy-diversity aspect, as previously outlined, we present a novel framework, named Category-based and Popularity-guided Video Game Recommender System (CPGRec), including a three-fold strategy. First, we introduce the Stringency-improved Game Connection module, a module aimed to improve accuracy by creating rigorous connections among games through cross-category associations. Second, we propose two components: (1) Connectivity-enhanced Neighbor Aggregation and (2) Popularity-guided Edges and Nodes Reweighting, which collectively contribute to the enhancement of diversity. The former component enhances the connectivity of the game graph to acquire neighbors from diverse categories with a low level of complexity. The latter component transforms popular game nodes into tools for spreading information about long-tail games by adjusting the weights of games based on their popularity and the weights of edges emanating from popular game nodes. Third, we design Combined Training with Negative-sample Score Reweighting, a comprehensive module that not only integrates the accuracy- and diversity-driven modules to create the ultimate representations of players and games but also introduces a novel method for calculating negative samples with extreme rating scores. This novel approach improves the model's capability to identify negative samples with high ratings to

recommend them less, and the frequency to recommend the negative samples with extremely low ratings, which are potentially long-tail games, thereby enhancing both accuracy and diversity in a balanced manner.

Our contributions can be summarized as follows:

- Our work represents the first effort within the field of video game recommendations, uniquely directed toward achieving a balance between accuracy and diversity.
- We propose a novel framework that enhances accuracy by creating strict interconnections among games based on cross-category associations, while enhancing diversity by promoting interactions among games across diverse categories and propagation between long-tail games and players.
- We design an innovative negative-sample score reweighting technique that encourages negative samples to manifest extreme predicted rating scores, aiming to improve accuracy and diversity in different extreme cases.
- We conduct a thorough evaluation to assess the effectiveness of our proposed method on a real-world video game dataset, demonstrating the superiority of our approach when compared to state-of-the-art methods in the field.

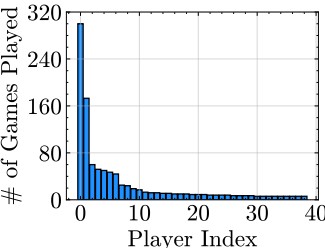

**Figure 1: Long-tail distribution on Steam dataset, marked by a significant presence of unpopular games, which players either show limited interest in or acquire but defer their engagement with.**

## 2 RELATED WORK

### 2.1 Video Game Recommendation

Game recommender systems have garnered increasing attention in academia, driven by the rapid expansion of the gaming industry. Early research has been particularly focused on methods based on Collaborative Filtering (CF) and Content-Based Filtering (CBF). Anwar *et al.* introduced a CF approach that treats games and users separately for personalized game recommendations [2]. BharathiPriya *et al.* combined CF and CBF techniques to estimate implicit rankings, considering factors such as playtime [3]. Similarly, Pérez-Marcos *et al.* proposed a hybrid system for video game recommendations, drawing inspiration from music domain methods to optimize playtime utilization [25].

Nowadays, there is a notable shift towards the exploration of innovative deep learning-based approaches for game recommendation research. These approaches have shown a tendency to outperform traditional methods that primarily rely on implicit feedback. For instance, Cheuque *et al.* conducted a comprehensive comparison study involving the traditional Factorization Machine (FM), the Deep Neural Network (DeepNN) model, and the hybrid DeepFM model [6]. Their findings, particularly when applied to the Steam dataset, demonstrated the superior performance of DeepNN and

DeepFM over the FM model. Additionally, the integration of GNNs into the realm of game recommendations has garnered significant attention. GNNs are well-suited for modeling complex relationships and dependencies within graph-structured data [11, 18, 45], making them a promising avenue for enhancing recommendation systems in the gaming domain. SCGRec, proposed by Yang *et al.*, is a novel game recommender system that leverages game contextualization and social connections using a GNN structure [40].

CPGRec diverges from the aforementioned studies in two critical aspects. (1) The current game recommendation works are mainly accuracy-driven approaches. While CPGRec also places considerable emphasis on accuracy, it is different from the SOTA SCGRec in its approach. Specifically, we harness CBF signals on game graphs with cross-category associations, as opposed to single-category associations, with the primary goal of establishing stricter connections among games to enhance accuracy. (2) Prior research endeavors do not address the issue of diversity, which can lead to monotonous recommended lists. In contrast, CPGRec takes into account both accuracy and diversity in its recommendation process.

## 2.2 Diversified Recommendation

Ziegler *et al.* [48] are among the early pioneers who introduced the concept of diversity into recommender systems. Since then, extensive research has been conducted to explore diversity in recommendation algorithms. For instance, Yin *et al.* conducted a comprehensive study to address diversity-related issues in session-based recommender systems [44]. DGCN, proposed by Zheng *et al.*, is a GNN-based method [18] that enhances diversity through neighbor selection, sample reweighting, and adversarial learning [47]. DD-Graph, introduced by Ye *et al.*, utilizes a diversified select operator to continuously update the user-item bipartite graph, ensuring the diversity of neighbors [42]. DGRec, proposed by Yang *et al.*, leverages a submodular function to select neighbors with higher diversity and incorporates multi-layer attention and loss reweighting mechanisms to extract amplified information from higher-order neighbors [41]. These works collectively contribute to advancing techniques for incorporating diversity in recommender systems.

CPGRec distinguishes itself from existing literature in four key aspects. (1) While previous works tend to compromise accuracy when enhancing diversity, our proposed model attempts to maintain accuracy by introducing a novel accuracy-driven module. (2) We directly enhance the graph structure to acquire neighbors with higher diversity by effectively utilizing the category information of games. This approach is distinct from dynamic methods based on neighbor selection or sample reweighting, which are more complex. (3) We introduce an innovative technique that adjusts edge and node weights within the player-game bipartite graph. This allows us to transform popular game nodes into instrumental nodes for disseminating messages from long-tail game nodes. (4) We enhance the calculation of the BPR Loss by considering the extreme rating scores of negative samples, thereby guiding the modeling of game embeddings and ultimately improving both accuracy and diversity. In contrast, the current loss reweighting mechanisms primarily rely on category information to improve only diversity.

## 3 PRELIMINARIES

This section introduces several fundamental preliminaries, including definitions of graphs and the problem formulation.

### 3.1 Graphs

*3.1.1* **DEFINITION(Game Graphs with Raw Connections)**. *In this study, we establish connections between games based on their category, which encompasses three attributes: genre, developer, and publisher, within a video game dataset. Utilizing this information, we establish the raw connection between games. For example, when there is an overlap between the genres of two games, they are connected. The principle is similarly applied to connections based on developer and publisher affiliations. By defining games connected through shared genre (g), developer (d), and publisher (p) attributes, we create three distinct game graphs denoted as $\mathcal{G}^g$, $\mathcal{G}^d$, and $\mathcal{G}^p$.*

*3.1.2* **DEFINITION(Player-game Bipartite Graph)**. *In the context of game recommendation, we are provided with a set of players denoted as $\mathcal{U} = \{u_1, u_2, \ldots, u_{|\mathcal{U}|}\}$ and games as $\mathcal{I} = \{i_1, i_2, \ldots, i_{|\mathcal{I}|}\}$. The set $\mathcal{I}(u)$ represents the collection of games that have been interacted with by the player u. To represent these historical player-game interactions in a graphical format, we construct the player-game bipartite graph denoted as $\mathcal{G} = (\mathcal{V}, \mathcal{E})$, where $\mathcal{V} = \mathcal{U} \cup \mathcal{I}$, and an edge connecting player u and game i is established if the player has engaged with the game.*

### 3.2 Problem Statement

The primary objective of this research is to develop a game recommender system with the goal of providing a top $K$ list of games $\mathcal{I}^{(u)} = \{i_1^{(u)}, i_2^{(u)}, \ldots, i_K^{(u)}\}$ that player $u$ has not yet interacted with. In addition, the research introduces a comprehensive task that combines both accuracy and diversity considerations. This task requires the recommended set of $K$ games to exhibit increased diversity while simultaneously minimizing the adverse impact on accuracy.

## 4 METHOD

### 4.1 Overview

As illustrated in Figure 2, the proposed framework, denoted as CPGRec, comprises four key components. These components are Stringency-improved Game Connection (SGC), Connectivity-enhanced Neighbor Aggregation (CNA), Popularity-guided Edges and Nodes Reweighting (PENR), and Combined Training with Negative-sample Score Reweighting (NSR).

SGC is the accuracy-driven module, while CNA and PENR collectively form the diversity-driven module. SGC and CNA both leverage category information to enhance accuracy and diversity, respectively. SGC achieves this by constructing high-stringency game graphs through cross-category associations, whereas CNA accomplishes it by constructing high-connectivity game graphs based on diverse-category neighbors. PENR utilizes the popularity information of games to enhance the likelihood of recommending long-tail games to users. Combined Training with NSR represents a comprehensive module that balances accuracy and diversity by combining the above three components and refining the BPR loss.

### 4.2 Stringency-improved Game Connection

In a typical game graph, the establishment of connections between two games is contingent upon their shared attributes, specifically within a defined category such as genre, developer, or publisher. In the existing investigations [5, 40, 47], it is apparent that the item graph is constructed based on raw connections. However, such a method of connection may not consistently yield a high degree

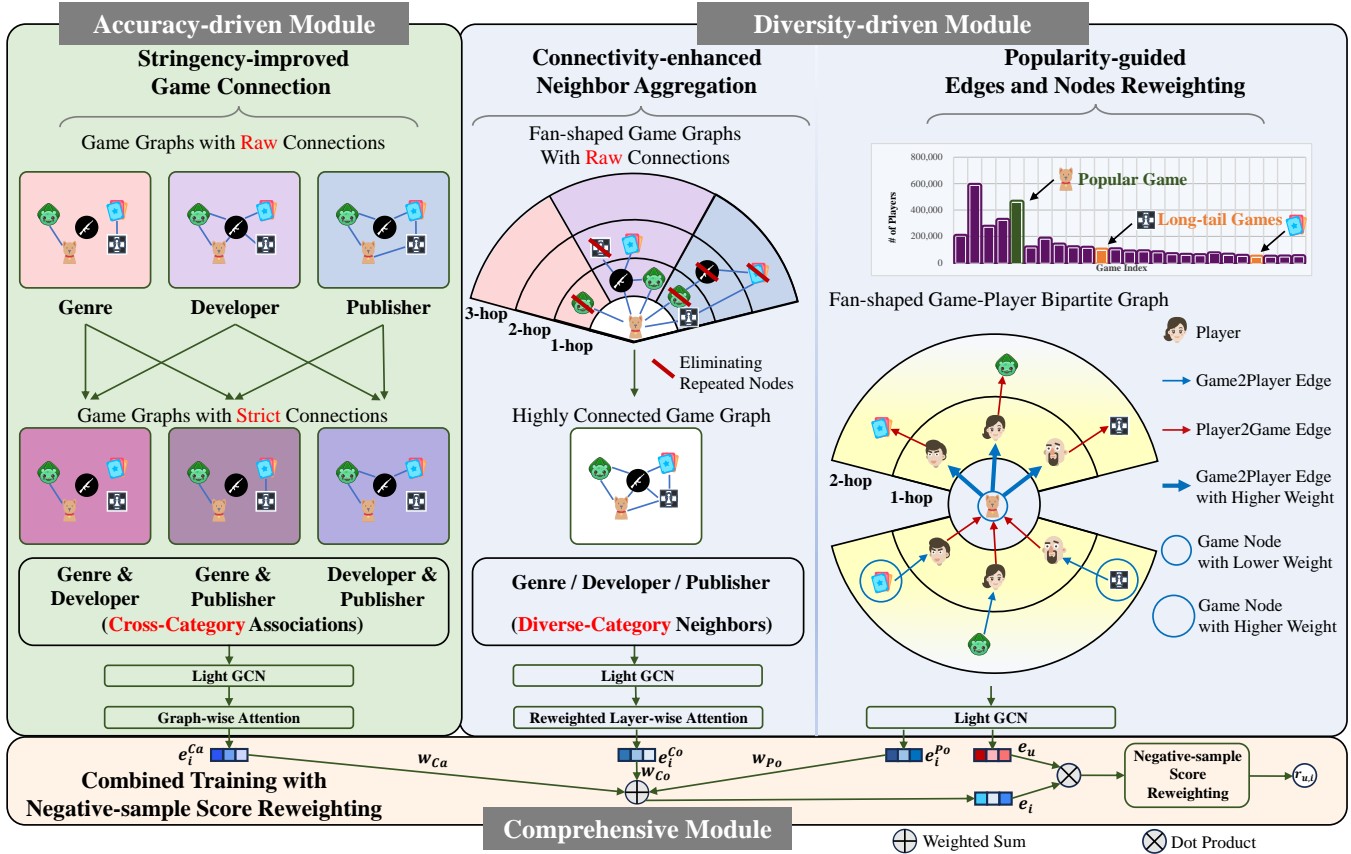

Figure 2: Illustration of the proposed framework of CPGRec. The left module is designed to emphasize accuracy, which is achieved through the implementation of SGC. The right module is tailored to prioritize diversity, which is realized by the joint utilization of CNA and PENR. SGC and CNA represent category-based components within the framework. SGC employs cross-category associations to establish strict connections between games, while CNA leverages diverse-category neighbors to enhance the connectivity of the game graph. The below module is a comprehensive unit that considers the balance between accuracy and diversity by Combined Training with NSR.

of reliability. This limitation arises from the fact that numerous games may concurrently share cross-category associations, offering a more multifaceted dimension of connectivity that, in turn, has the potential to establish stricter interconnections among games. As an illustrative example from the Steam dataset, as depicted in Figure 3, it becomes evident that the Game Graphs with strict connections exhibit a notably lower number of edges, specifically 6,236, for the combined categories of genre & developer. In contrast, the Game Graphs with raw connections contain a significantly higher number of edges, with 624,236 and 9,630 for the genre and developer categories, respectively. This discrepancy underscores the more rigorous and selective edge connections present in the Game Graphs with strict connections.

We create game graphs with strict connections by maintaining connections only between games that share a minimum of two categories. Specifically, we preserve connections between games characterized by both the same genre ($g$) and developer ($d$), thereby establishing a game graph denoted as $\mathcal{G}^{g\&d}$, exemplifying cross-category associations. Similarly, we derive game graphs denoted as $\mathcal{G}^{g\&p}$ and $\mathcal{G}^{d\&p}$, where $p$ refers to the category of publisher ($p$).

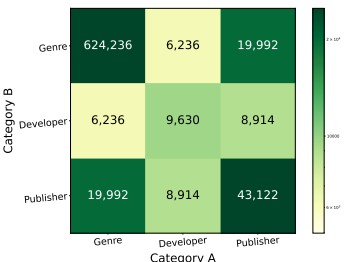

Figure 3: Number of edges on Steam game graphs, where the diagonal elements represent the counts of edges in the game graphs with raw connections, while the non-diagonal elements correspond to the counts of edges in the game graphs with strict connections.

Subsequently, we utilize the LightGCN [13] and attention mechanism [34] to learn the embedding of game $i$ in this module as follows:

$$e_i^{Ca} = Graphwise\_Attention(e_i^{g\&d}, e_i^{g\&p}, e_i^{d\&p}), \quad (1)$$

where $e_i^{g\&d}, e_i^{g\&p}, e_i^{d\&p}$ indicate the outputs of LightGCN from $\mathcal{G}^{g\&d}, \mathcal{G}^{g\&p}$ and $\mathcal{G}^{d\&p}$, respectively.

## 4.3 Connectivity-enhanced Neighbor Aggregation

In order to foster diversity, we enhance the interconnectedness of the game graph by facilitating the propagation and aggregation of messages between games of different categories.

We establish a game graph denoted as $\mathcal{G}^{Co}$ in which neighboring games are constrained to share a solitary common category. Importantly, it should be emphasized that these neighboring games may substantially diverge in terms of other categorical attributes. For instance, it is entirely feasible for two adjacent games to share the same genre while markedly differing in their developer and publisher affiliations. Furthermore, it is essential to highlight that the variance in the categories of a game's neighbors can be quite substantial, even if they are merely separated by a single node. For example, in the Steam dataset, the neighbors associated with the node representing the video game "Virtual DJ" on the graph $\mathcal{G}^{Co}$ traverse a total of 64 distinct categories, comprising 18 genres, 29 developers, and 17 publishers. Remarkably, these nodes are interconnected with the game "Virtual DJ" acting as the intermediary connecting them. This observed phenomenon serves as compelling evidence that $\mathcal{G}^{Co}$ can be regarded as a highly connected game graph. This high level of connectivity facilitates the effective propagation of messages across disparate categories.

To enhance the cross-category message interaction, multi-layer LightGCN is applied to propagate and aggregate messages within $\mathcal{G}^{Co}$. We also use an attention mechanism to assign appropriate weights to the embeddings generated by various layers, mitigating the potential issue of over-smoothing. Additionally, a mandatory layer-wise reweighting parameter, shown as follows, is incorporated to further fine-tune the message interactions in the network:

$$w_l = 1 - (k - l)\beta, \tag{2}$$

where $w_l$ is the reweighting parameter for the $l$-th layer, $k$ is the number of LightGCN layers, and $\beta$ is the decay parameter that assigns relatively greater weight to the embeddings produced by deeper layers. Specifically, $w_l$ serves the purpose of weighting the game embedding of $l$-th layer to enhance the significance of embeddings originating from deeper layers, which are inherently more likely to carry messages from more distant and diverse games. Here, a deeper layer is assigned a higher value for $w_l$. Thus, the embedding of game $i$ in this module is calculated as:

$$e_i^{Co} = Layerwise\_Attention(w_1 e_i^{(1)}, w_2 e_i^{(2)}, ..., w_k e_i^{(k)}), \tag{3}$$

where $e_i^{(l)}$ is the output embedding of game $i$ from $l$-th layer.

## 4.4 Popularity-guided Edges and Nodes Reweighting

The bipartite graph $\mathcal{G}$ representing the relationship between players and games offers a direct representation of historical player-game interactions within a structured network. This graph encapsulates potent collaborative signals. Nevertheless, it inherently reveals a notable challenge associated with the lower popularity of long-tail games [1, 31], manifesting in their limited connectivity within the graph. This situation poses obstacles to message interactions, involving both propagation and aggregation, between long-tail games and a broader player pool. Therefore, these long-tail games face impediments in terms of receiving recommendations.

To empower long-tail games during the message propagation process, we leverage the extensive connectivity of popular games by enabling them to propagate messages with significantly greater impact. Specifically, popular game nodes are repurposed as vehicles for propagating messages of long-tail games. This is accomplished by amplifying the weight of edges emanating from popular game nodes, while concurrently diminishing the weight of nodes (i.e., weight of embeddings) of popular game nodes themselves. The former adjustment serves to bolster the messaging propagation capabilities of popular game nodes, even though it may seem counterintuitive to our objective. Nevertheless, the latter modification is employed to counterbalance the inherent advantage enjoyed by popular games owing to their higher connectivity and the higher edge weight mentioned above. This ensures that the information predominantly propagated originates from long-tail games, aligning with our objective of prioritizing these less popular games.

We designate games with player counts falling within the top 20% as "popular games" whereas those falling within the bottom 20% as "long-tail games". The set of popular games is denoted as $\mathcal{I}_{hot}$, while the set of long-tail games is represented as $\mathcal{I}_{cold}$. For the former category, we introduce a parameter $\theta_e^{hot}$ to enhance the weight of edges emanating from popular games, and we utilize $\theta_n^{hot}$ as the weight of nodes for these popular games. Conversely, for the latter category, we employ a higher value of $\theta_n^{cold}$ to emphasize the weights of nodes for long-tail games, further ensuring their influence. An edge weight mapping $\Theta_e(\cdot)$ is defined as:

$$\Theta_e(i) = \begin{cases} \theta_e^{hot} & i \in \mathcal{I}_{hot} \\ 1 & i \notin \mathcal{I}_{hot} \end{cases}, \tag{4}$$

and similarly a node weight mapping $\Theta_n(\cdot)$ is defined as:

$$\Theta_n(i) = \begin{cases} \theta_n^{hot} & i \in \mathcal{I}_{hot} \\ 1 & i \notin \mathcal{I}_{hot} \cup \mathcal{I}_{cold} \\ \theta_n^{cold} & i \in \mathcal{I}_{cold} \end{cases}. \tag{5}$$

The graph convolution layer incorporating the above reweighting mechanism is defined as:

$$e_u^{(l+1)} = \frac{1}{\sqrt{|\mathcal{N}_u|}\sqrt{|\mathcal{N}_u|}} e_u^{(l)} + \sum_{i \in \mathcal{N}_u} \frac{\Theta_e(i)\Theta_n(i)}{\sqrt{|\mathcal{N}_u|}\sqrt{|\mathcal{N}_i|}} e_i^{(l)}, \tag{6}$$

$$e_{(i)}^{(l+1)} = \frac{\Theta_n(i)}{\sqrt{|\mathcal{N}_i|}\sqrt{|\mathcal{N}_i|}} e_i^{(l)} + \sum_{u \in \mathcal{N}_i} \frac{1}{\sqrt{|\mathcal{N}_i|}\sqrt{|\mathcal{N}_u|}} e_u^{(l)}, \tag{7}$$

where $\mathcal{N}_i$ and $\mathcal{N}_u$ represents the neighbor set of game $i$ and player $u$, respectively. $e_i^{(l)}, e_u^{(l)}$ are the output embeddings of game $i$ and player $u$ from the $l$-th layer of LightGCN.

After $k$ layers, the embeddings of player $u$ and game $i$ in this module are calculated as:

$$e_u^{Po} = e_u^{(k)}, e_i^{Po} = e_i^{(k)}. \tag{8}$$

## 4.5 Combined Training with Negative-sample Score Reweighting

The challenge of delivering players a satisfying gaming experience lies in striking an appropriate balance between accuracy and diversity. The first and last two modules proposed above constitute the accuracy-driven module and diversity-driven module of our framework, respectively. To synthetically incorporate these considerations, CPGRec employs a weighted sum approach under the

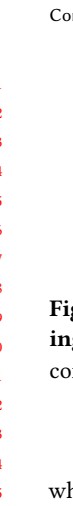
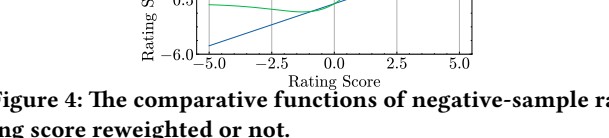

**Figure 4: The comparative functions of negative-sample rating score reweighted or not.**

control of an adjustable parameter $\gamma$ defined as:

$$e_u = e_u^{Po}, e_i = w_{Ca}e_i^{Ca} + w_{Co}e_i^{Co} + w_{Po}e_i^{Po}, \tag{9}$$
$$w_{Ca} + w_{Co} + w_{Po} = 1,$$

where $w_{Ca}, w_{Co}, w_{Po}$ represent the weights assigned to the embeddings $e_i^{Ca}, e_i^{Co}, e_i^{Po}$ of game $i$, respectively. $e_i$ and $e_u$ are the final embedding of game $i$ and player $u$. By adjusting different weights, we can adaptively apply the proposed model to fulfill various requirements, which will be detailed in Section 5.4.

In this module, CPGRec further practices the idea of harmonizing accuracy and diversity during the training process by refining the methodology employed to calculate the rating scores of negative samples that exhibit extreme scores, which is defined as:

$$\widetilde{L}_{BPR} = -\sum_{\substack{u \in \mathcal{U}, i \in \mathcal{I}(u), \\ j \notin \mathcal{I}(u)}} \log \sigma(r_{u,i} - \widetilde{r}_{u,j}) + \lambda \|\Theta\|_2^2, \tag{10}$$

where $\sigma(\cdot)$ is the sigmoid function, $r_{u,i}$ is the rating score between player $u$ and game $i$ calculated by their inner product as follows:

$$r_{u,i} = e_u \cdot e_i, \tag{11}$$

and the negative-sample score reweighting is:

$$\widetilde{r}_{u,j} = m \cdot \sigma(r_{u,j}) \cdot r_{u,j}, \tag{12}$$

where $\widetilde{r}_{u,j}$ refines the calculation of the rating score of a negative sample $j$ in terms of both accuracy and diversity. $m$ is a hyper-parameter for controlling the intensity of reweighting.

Figure 4 provides an intuitive depiction of the effects of negative-sample score reweighting. As we can observe, this reweighting process predominantly impacts negative samples that exhibit significantly high or low scores, while those with moderate scores remain largely unaffected. These extreme values can be utilized to improve accuracy and diversity during the training process.

From the perspective of **enhancing accuracy**, a negative sample predicted with a high rating score could be deceptive, i.e. it can easily be mistaken for a positive sample, even with a high rating. Thus, the elevated loss incurred through negative-sample score reweighting compels an improvement in the model's recognition capabilities, with the aim of achieving more accurate predictions in such cases.

From the perspective of **promoting diversity**, a low score assigned to a negative sample implies a substantial dissimilarity between that sample and players in terms of their embeddings. This could be indicative of the limited interactions due to the poor exposure of the game to players, suggesting that it is potentially a long-tail game. By increasing the rating score, this module mitigates the penalty that negatively affects games with such characteristics, thereby enhancing their chances of being recommended.

# 5 EXPERIMENTS

## 5.1 Experimental Setup

*5.1.1 Dataset.* To evaluate the effectiveness of CPGRec, we conducted experiments on the Steam Dataset contributed by Mark et.al. [24] and Yang et.al. [40], which provides comprehensive data about both players and games. Notably, CPGRec utilized the categories of genre, developer, and publisher of games from this dataset. To ensure the quality of data, 5-core setting was adopted to filter out the players with fewer than 5 game interactions, while retaining games with relatively low player engagement. These long-tail games are of particular interest as they represent the focus of our recommendation efforts. The statistics of the filtered Steam Dataset are shown in Table 1.

**Table 1: Statistics of the filtered Steam dataset.**

| Dataset | Steam |
|---|---|
| # Users | 3,908,744 |
| # Games | 2,675 |
| # Interactions | 95,208,806 |
| # Genre | 22 |
| # Developer | 1,170 |
| # Publisher | 688 |

The dataset was partitioned as follows: 80% of the data was randomly allocated for training purposes, with an additional 10% designated for validation and another 10% reserved for testing. The validation sets were utilized in hyper-parameter tuning during the experimentation process.

*5.1.2 Evaluation Metrics.* Inspired by previous works [7, 37, 40, 41], our evaluation metrics include the accuracy-focuses measures, such as NDCG@$K$, Recall@$K$, Hit@$K$ and Precision@$K$. Additionally, we consider a diversity metric, Coverage@$K$, where $K$ is explored across the set {5,10}. Specifically, Coverage@$K$ is calculated by summing the number of distinct genres, developers, and publishers corresponding to the top $K$ games in the recommended list.

*5.1.3 Baselines.* For a comprehensive evaluation of CPGRec, we conducted comparisons with several representative recommender models. In terms of accuracy, we compared CPGRec with two models: the simplified yet efficient LightGCN [13], and SCGRec [40], which represents the state-of-the-art accuracy-driven game recommendations. Concerning diversity, we assessed CPGRec against three noteworthy GNN-based diversity-driven models: DDGraph [42], DGCN [47], and DGRec [41].

*5.1.4 Implementation Details.* In the following experiments, we implement CPGRec using Pytorch and the DGL, which is tailored for deep learning on graphs. More details, we employ the Adam optimizer for training. The learning rate is set to 0.03, and the batch size is 1024. Following the configuration of SCGRec and DGCN, we set the embedding size to a fixed value of 32. The decay parameter $\beta$ is set to 0.1, and the intensity parameter $m$ is set to 6.5. To prevent overfitting, we apply an early stop strategy during training. We will also provide detailed tuning of hyperparameters in the following sections.

## 5.2 Performance Evaluation

A comprehensive comparison of CPGRec with other baselines is reported in Table 2. The best and second-best results are highlighted

**Table 2: Performance comparison of different accuracy- and diveristy-driven recommender systems.**

| | Method | NDCG | | Recall | | Hit | | Precision | | Coverage | |
|---|---|---|---|---|---|---|---|---|---|---|---|
| | | @5 | @10 | @5 | @10 | @5 | @10 | @5 | @10 | @5 | @10 |
| Accuracy-driven Methods | LightGCN | 0.1861 | 0.2100 | 0.2452 | 0.3174 | 0.2849 | 0.3784 | 0.0637 | 0.0447 | 7.6572 | 12.8169 |
| | SCGRec | 0.4351 | 0.4660 | 0.5385 | 0.6311 | 0.6519 | 0.7535 | 0.1508 | 0.0969 | 7.9477 | 12.9688 |
| | **Accuracy-focused CPGRec** | **0.4796** | **0.5000** | **0.5746** | **0.6387** | **0.6983** | **0.7659** | **0.1625** | **0.0989** | **8.5092** | **15.5399** |
| Diversity-driven Methods | DDGraph | 0.3997 | 0.4298 | 0.4949 | 0.5883 | 0.6059 | 0.7094 | 0.1399 | 0.0894 | 8.1288 | 13.8425 |
| | DGCN | 0.3732 | 0.4025 | 0.4536 | 0.5523 | 0.6056 | 0.7123 | 0.1129 | 0.0806 | 8.1913 | 14.0753 |
| | DGRec | 0.3546 | 0.3982 | 0.4293 | 0.5434 | 0.5791 | 0.7126 | 0.1041 | 0.0786 | 8.8520 | 14.2742 |
| | **Diversity-focused CPGRec** | **0.4285** | **0.4547** | **0.5168** | **0.5990** | **0.6390** | **0.7292** | **0.1469** | **0.0922** | **10.1377** | **17.6409** |
| Balance-driven Method | **CPGRec (trade-off framework)** | 0.4320 | 0.4582 | 0.5223 | 0.6044 | 0.6449 | 0.7342 | 0.1489 | 0.0934 | 9.253 | 16.7079 |

in bold and underlined, respectively. Based on these experimental results, the following observations can be made:

- CPGRec surpasses all diversity-driven baselines w.r.t Coverage. In terms of accuracy, CPGRec ranks second, trailing only behind SCGRec, the SOTA accuracy-driven model. This highlights that CPGRec achieves superior diversity while maintaining a high level of accuracy compared to other diversity-driven models.
- Accuracy-focused CPGRec (CPGRec w/o Diversity-driven Module) outperforms all current accuracy-driven methods across all metrics, while Diversity-focused CPGRec (CPGRec w/o Accuracy-driven Module) outperforms all current diversity-driven methods across all metrics. This underscores the SOTA performance of both configurations.
- While SCGRec consistently performs the best in all accuracy metrics, its Coverage falls significantly behind three diversity-driven models, including CPGRec. This suggests that accuracy-driven models tend to prioritize accuracy at the expense of diversity, leading to a substantial imbalance between the two aspects. Such an imbalance can have a negative impact on the gaming experience of players.
- Category-based CBF signals play a crucial role in achieving higher accuracy. Both SCGRec and CPGRec, which propagate and aggregate messages on game graphs with category-based similarity (representing the CBF signals), achieve the highest levels of accuracy. This highlights the importance of leveraging category information to improve accuracy in game recommendation systems.
- Leveraging information from higher-order neighbors emerges as a viable approach to enhance diversity. Both DGRec and CPGRec, representing the highest and slightly lower levels of diversity, respectively, enable nodes to access information from high-order neighbors through the stacking of multiple GNN layers. Additionally, CPGRec enhances the importance of embeddings associated with high-order neighbors through its layer-wise reweighting design, leading to significantly improved diversity.

### 5.3 Ablation Study

In this section, we perform an ablation study on CPGRec by removing each of the four modules. We have the following observations based on experiment results shown in Table 3:

- Removing SGC from the accuracy-driven module leads to a variant that achieves the highest diversity at the cost of an overall decline in accuracy. This outcome aligns with the accuracy-diversity trade-off dilemma: in the presence of a

sole diversity-driven module, accuracy tends to be sacrificed in favor of heightened diversity.

- The removal of CNA results in an observable, albeit anticipated, reduction in diversity, along with a modest decline in accuracy. This outcome underscores the pivotal role of CNA, as it not only substantively contributes to diversity enhancement but also marginally improves accuracy by introducing valuable information from higher-order neighbors. The evidence for the efficacy of using high-order neighbor information to enhance accuracy has been substantiated in a previous study [6]. Thus, the incorporation of high-order neighbors with diverse categories aligns with our commitment to balancing accuracy and diversity.
- The variant that lacks PENR exhibits superior performance in accuracy-based metrics but experiences a slight decline in Coverage. This suggests that PENR sacrifices a greater degree of accuracy compared to CNA. This phenomenon may be attributed to the inherent trade-off associated with PENR which necessitates a transformation of the player-game bipartite graph. This transformation, while intended to enhance diversity, inadvertently introduces noise, rendering it more challenging for the model to discern vital historical interaction information between players and games. Furthermore, when compared to CNA, PENR demonstrates a less pronounced increase in Coverage. This finding highlights the significance of leveraging information from high-order neighbors to improve diversity in the recommendation process.
- The variant without NSR exhibits the poorest performance in terms of both accuracy and diversity across all metrics. This underscores the significance of NSR, as it plays a pivotal role in enhancing accuracy by increasing the losses linked to negative samples with high rating scores and fostering diversity by enhancing potential rating scores linked to potential long-tail samples.

### 5.4 Parameter Sensitivity

In this section, we explore the impact of various hyperparameters used in CPGRec in order to achieve a more balanced trade-off between accuracy and diversity.

*5.4.1 Sensitivity Analysis on $\theta_e^{hot}, \theta_n^{hot}, \theta_n^{cold}$.* As shown in Equation 4, $\theta_e^{hot}$ is the enhanced weight of edges from popular game nodes to players on the bipartite graph $\mathcal{G}$, while $\theta_n^{hot}, \theta_n^{cold}$ in Equation 5 correspond to the node weights of popular games and long-tail games, respectively. Figure 5 illustrates the impact of varying these values on both recommendation accuracy and diversity.

**Table 3: Ablation study. We show CPGRec's performance when removing each of the components.**

| Method | Recall@5 | Recall@10 | Hit@5 | Hit@10 | Precision@5 | Precision@10 | Coverage@5 | Coverage@10 |
|---|---|---|---|---|---|---|---|---|
| **CPGRec** (**trade-off** framework) | 0.5223 | 0.6044 | 0.6449 | 0.7342 | 0.1489 | 0.0934 | 9.2532 | 16.7079 |
| w/o SGC (accuracy-driven component) | 0.5168 | 0.5990 | 0.6390 | 0.7292 | 0.1469 | 0.0922 | **10.1377** | **17.6409** |
| w/o CNA (1st diversity-driven component) | 0.5165 | 0.6019 | 0.6382 | 0.7315 | 0.1475 | 0.0930 | 8.1179 | 14.7422 |
| w/o PENR (2nd diversity-driven component) | **0.5689** | **0.6281** | **0.6907** | **0.7546** | **0.1601** | **0.0965** | 9.0271 | 16.5351 |
| w/o NSR (comprehensive component) | 0.4907 | 0.5881 | 0.6080 | 0.7149 | 0.1405 | 0.0909 | 8.0430 | 13.5161 |

Concerning $\theta_e^{hot}$, a consistent reduction in accuracy is observed as it decreases. Conversely, the increase in diversity is more pronounced as messages from long-tail games propagate more extensively, either directly or indirectly, facilitated by edges assigned higher weights. However, the Coverage@5 metric levels off when $\theta_e^{hot}$ increases from 40 to 50, indicating that increasing the edge weight from popular games does not always guarantee an improvement in diversity. Excessively high weights for these edges can lead to a disregard for the significance of edges originating from non-popular games, given their relatively lower weight. This situation mirrors that of long-tail games and ultimately results in a decline in diversity.

For $\theta_n^{hot}$, reducing the node weight of popular game nodes limits their significance, leading to a decrease in accuracy as it diminishes. The improvement in diversity reflects the accuracy-diversity trade-off dilemma. However, excessively reducing the weights of popular games places them in a situation similar to that of long-tail games, ultimately constraining diversity.

Regarding $\theta_n^{cold}$, enhancing the node weight of long-tail games gradually from 1 to 5 results in the expected outcome: the model improves diversity significantly at the cost of accuracy by assigning higher weights to long-tail games. However, it is important to note that when long-tail games are given excessively high weights (e.g., 7, 9), both accuracy and diversity decrease. This is due to long-tail games taking the place of originally popular games, resulting in a significant portion of players' recommended lists being occupied by less likely-to-be-clicked long-tail games.

*5.4.2 Sensitivity Analysis on* $w_{Ca}, w_{Co}, w_{Po}$. Expressed in Equation 9, $w_{Ca}$ represents the weight of the accuracy-driven module, while the combined $w_{Co}$ and $w_{Po}$ signify the importance of the diversity-driven module. In our study, we showcase CPGRec's capacity to strike a balance between accuracy and diversity by tuning the parameters $w_{Ca}$ and $w_{Co}$. We determine $w_{Co}$ to signify the weight of the diversity-driven module based on the findings of ablation experiments, which suggest that, compared to PENR, CNA is more crucial for diversity. We set $w_{Po} = 1 - w_{Ca} - w_{Co}$ to maintain the balance. The sensitivity results are presented in Figure 6.

As the parameter $w_{Ca}$ gradually increases, CPGRec consistently improves its accuracy performance. Regarding diversity, as measured by Coverage@5, CPGRec shows fluctuations but generally maintains around 8.5. This behavior contrasts with the typical accuracy-diversity dilemma, where diversity often sharply decreases as accuracy increases. CPGRec effectively manages to maintain diversity while enhancing accuracy. Conversely, when increasing the parameter $w_{Co}$, CPGRec initially experiences a significant rise in diversity before stabilizing at a high level. Notably, unlike the expected trend, CPGRec's accuracy does not continuously decline but sacrifices some degree of stability while generally remaining at an acceptable level.

These experimental results emphasize that the relationship between accuracy and diversity is not always a straightforward trade-off. Instead, it can exhibit a more intricate interplay, where enhancing one aspect might lead to fluctuations in the other's stability. This observation aligns with previous research, as proposed by [44], and underscores CPGRec's robust ability to balance accuracy and diversity effectively.

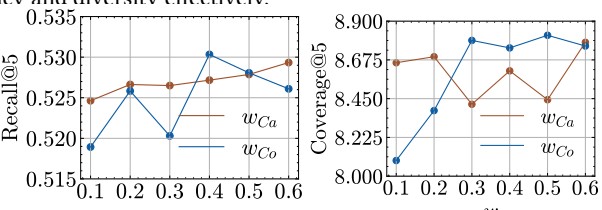

**Figure 6: Parameter sensitivity on $w_{Ca}$ and $w_{Co}$.**

## 6 CONCLUSION

This study introduces an innovative model, CPGRec, explicitly engineered to harness the potential of two factors: game categories and popularity. The primary objective is to strike a balance between accuracy and diversity in video game recommendation. In addition, a novel negative-sample score reweighting technique is incorporated into the loss function to enhance both accuracy and diversity. Our empirical results substantiate the superiority of CPGRec, surpassing recent diversity-driven recommender systems in both accuracy and diversity. Regarding accuracy, CPGRec closely approaches the performance of the SOTA accuracy-centric game recommendation system, trailing by only a small margin.

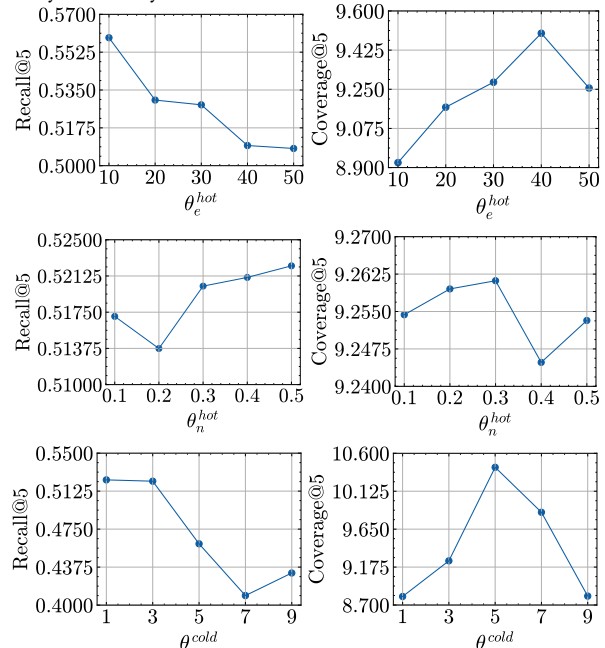

**Figure 5: Parameter sensitivity on $\theta_e^{hot}, \theta_n^{hot}$, and $\theta_n^{cold}$.**

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

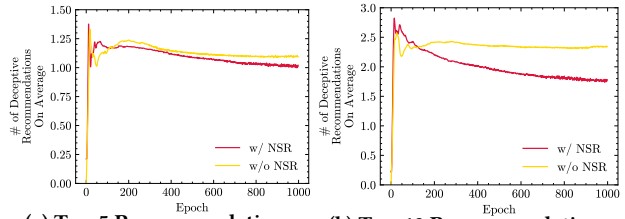

(a) Top-5 Recommendations    (b) Top-10 Recommendations

**Figure 7: The average number of deceptive games recommended to all players in both the (a) Top-5 and (b) Top-10 recommendations.**

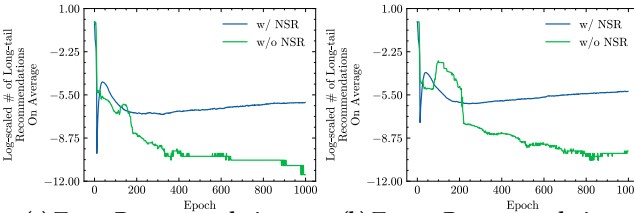

(a) Top-5 Recommendations    (b) Top-10 Recommendations

**Figure 8: The average number of long-tail games recommended to all players in both the (a) Top-5 and (b) Top-10 recommendations.**

## A  CASE STUDY

In this section, we delve into a case study dedicated to the comprehensive module of CPGRec. The primary objective is to gain a thorough understanding of how NSR can effectively improve both accuracy and diversity within the system.

### A.1  Case Study on Accuracy

*A.1.1  Experimental Setup.* In terms of accuracy, given limited computational resources and a substantial player base, we employ a random sampling approach, selecting one-fourth of all players as a representative sample. The primary goal is to identify the most deceptive negative sample games. To accomplish this, we train the model without introducing the NSR technique and identify the top 10 negative sample games with the highest occurrence frequency among the Top-5 recommendations for the sampled players. These games are identified as the most deceptive negative sample games.

Subsequently, we conduct a comparative analysis, observing the average recommendation frequencies of these highly deceptive negative sample games within both the Top-5 and Top-10 recommendations for models trained with and without the consideration of NSR. Importantly, this analysis encompasses all players, not restricted to the sample players. To mitigate the potential impact of random fluctuations in the experimental results, we conduct the experiments using five different seeds and calculate their average values to derive the final comparison.

*A.1.2  Analysis.* The experimental results, as illustrated in Figure 7a for Top-5 recommendations and Figure 7b for Top-10 recommendations, provide the following key observation.

- Upon the introduction of the NSR technique, there is a noticeable reduction in the recommendation frequencies for deceptive games. This indicates that NSR improves the model's ability to identify these games, rendering the model less susceptible to their misleading influence.
- Throughout the training process, the model's metrics display initial instability with fluctuating patterns, characterized by an initial increase, followed by a subsequent decrease, and then a second increase. Subsequently, the metrics enter a sustained descending phase until the training concludes. The initial stage corresponds to significant changes in the model's embeddings, which contributes to the fluctuating recommendation results. The latter phase signifies that the model has established a stable direction for embedding modeling. Notably, with the introduction of NSR, the model reaches this stable phase earlier, signifying that NSR facilitates the

model in identifying the correct direction for embedding modeling.
- Irrespective of the presence of NSR, the model's metrics exhibit nearly identical patterns in the initial training stage. However, the model incorporating NSR progressively reduces its recommendations for deceptive games during the training process, underscoring NSR's role in enhancing the model's recognition capabilities over time.

### A.2  Case Study on Diversity

*A.2.1  Experimental Setup.* Regarding diversity, our analysis focuses on the number of recommendations for all long-tail games, as defined in Section 4.4, within the Top-5 and Top-10 recommendations. This evaluation is conducted on average across all players and is performed across various training epochs to assess how the NSR technique guides the model in enhancing diversity. We make comparisons between scenarios with and without NSR to facilitate a more discernible comparison. To enhance the clarity of this analysis, we introduce the use of a logarithmic scale.

*A.2.2  Analysis.* The experimental results, as depicted in Figure 8a for Top-5 recommendations and Figure 8b for Top-10 recommendations, offer the following insightful observations.

- The introduction of the NSR technique leads to an increased frequency of recommendations for long-tail games throughout the training process. This underscores the efficacy of NSR in enhancing diversity by augmenting the number of recommendations for long-tail games.
- Irrespective of the presence of NSR, the recommendation frequencies for long-tail games follow similar developmental trends during the training process: an initial decrease, followed by an increase, and subsequently another decrease. The metrics reach their peak in the early stages of training, as the training process has not yet exposed the limitations of long-tail games. After these initial fluctuations, the metrics gradually enter a relatively stable phase.
- In models without NSR, the metrics exhibit a distinct overall decline, attributed to the emphasis on accuracy improvement driven by loss function optimization. However, with the introduction of the NSR module, the recommendation frequencies for long-tail games are consistently maintained at a significantly higher level, displaying a slight upward trend during the training process. This signifies that NSR empowers long-tail games to receive more stable and frequent recommendations, contributing to enhanced diversity.

