# OpenReview forum: "Category-based and Popularity-guided Video Game Recommendation: A Balance-oriented Framework"
_ACM.org/TheWebConf/2024/Conference — TheWebConf24 Oral_

### Official Review · Reviewer_QhsD · 2023-11-14

**Novelty:** 5
**Technical Quality:** 4

**Review:**

This paper investigates category-based and popularity-guided video game recommendations for balancing the trade-off between recommendation accuracy and diversity.
The paper studies a very interesting problem, since video game recommender systems are a growing research field, and also investigating diversity in recommendations is still a very relevant topic.
The authors also provide a nice approach by proposing three components to address the accuracy - diversity trade-off.
The evaluation results on the Steam dataset shows that their approach is capable of providing good results with respect to both dimensions, i.e., accuracy and diversity.

However, there are some weak points in the paper that need to be addressed before it can be published and presented at the WebConf:
1) In my opinion the author mix up the concepts of diversity and novelty/serendipity (see question below)
2) Figure 1 shows the number of games played for the users, but the authors state that this plot shows that there are many long-tail items. Thus, the authors mix up the concepts of long-tail items and varying user history lengths
3) There are issues with the definition of the 5-core evaluation setting (see Question 3)
4) There are issues with the choice of the evaluation metrics (see Question 4+5)
5) No future work ideas are provided

**Questions:**

1) In the introduction, you relate the lack of diversity in recommendation lists to the issue of filter bubbles and echo chambers. But isn't this more a problem of a lack of novelty and serendipity? Please elaborate more on this
2) How does Figure 1 illustrate the long-tail distribution of items?
3) In Section 5.1., how is the 5-core setting applied? On the items and on the users? If it is also applied on the items, how does this impact long-tail items? Aren't they removed then?
4) If this work is about diversity, why you do not include any diversity metrics (e.g., the intra-list diversity of recommended items)?
5) Why don't you calculate the coverage separately for genres, developers and publishers? I think this would be more in line with the original definition of recommendation coverage.

**Ethics Review Description:**

-

**Reviewer Confidence:**

3: The reviewer is confident but not certain that the evaluation is correct

**Scope:**

3: The work is somewhat relevant to the Web and to the track, and is of narrow interest to a sub-community

---

### Official Review · Reviewer_V4y1 · 2023-11-22

**Novelty:** 5
**Technical Quality:** 5

**Review:**

The paper introduces a comprehensive framework for video game recommendation that incorporates category-based considerations, popularity guidance, and a balance between accuracy and coverage. The experimental results suggest the effectiveness of the proposed modules: Stringency-improved Game Connection (SGC), Connectivity-enhanced Neighbor Aggregation (CNA), Popularity-guided Edges and Nodes Reweighting (PENR), and Combined Training with Negative-sample Score Reweighting (NSR). Furthermore, the authors generously provide code to enhance comprehension. However, some ambiguities persist, particularly in relation to the module functionalities and metric definitions, prompting lingering questions that require clarification.

**Questions:**

1. The coverage metric falls within the domain of aggregate diversity. In research, diversity is often categorized into two groups: individual diversity and aggregate diversity. However, the Introduction section's discussion contains some inaccuracies and inconsistencies that need clarification, like “While these methods have indeed made significant strides in enhancing accuracy, they often overlook the crucial aspect of recommendation list diversity. This oversight raises concerns about the echo chamber/filter bubble effect [12, 19, 22, 38], where users are confined to familiar games. Such confinement limits their exposure to diverse perspectives and has the potential to reinforce biases.”.

2. In the proposed Stringency-improved Game Connection (SGC) module, game graphs with stringent connections are employed instead of raw connections. However, the ablation study only implements the model without SGC. Additionally, I am curious about the experimental results of CPGRec, where strict connections are replaced with raw connections in the ablation study.

3. I have reservations regarding the diversity motivation behind the Connectivity-enhanced Neighbor Aggregation (CNA). It appears to combine three game graphs with raw connections from genre, developer, and publisher perspectives. Moreover, in the ablation study, excluding CNA results in decreases in both accuracy and coverage. I believe the CNA module is more inclined to aggregate information from raw connections, which can supplement with the information from strict connections in SGC. Consequently, the diversity motivation in CNA does not seem apparent in this context.

4. The coverage metric's definition in the paper diverges from the common understanding, deviating from the typical coverage definition that counts how many distinct items appear in the top-K recommendations [1]. Instead, the paper calculates coverage by summing the number of distinct genres, developers, and publishers corresponding to the top K games in the recommended list. This uncommon definition may introduce bias, particularly for other baselines that do not utilize side information such as genre, developer, and publisher. I advocate for adopting the conventional coverage definition commonly used in research for a fair and standardized evaluation.
[1] Siyi Liu and Yujia Zheng. 2020. Long-tail Session-based Recommendation. In Proceedings of the 14th ACM Conference on Recommender Systems (RecSys '20). Association for Computing Machinery, New York, NY, USA, 509–514. https://doi.org/10.1145/3383313.3412222

**Reviewer Confidence:**

4: The reviewer is certain that the evaluation is correct and very familiar with the relevant literature

**Scope:**

4: The work is relevant to the Web and to the track, and is of broad interest to the community

---

### Official Review · Reviewer_5sGQ · 2023-11-27

**Novelty:** 3
**Technical Quality:** 3

**Review:**

This paper approaches the problem of game recommendation, considering both accuracy and diversity, with a three-module neural network.
The first module considers the Cross-Category Association in graph learning. The second module connects neighbors with diverse categories, thereby enriching recommendation diversity. The third module combines the above two modules.


The strengths of this paper are as follows.
1. The studied problem of diversified recommendation is of great importance.
2. The proposed method is suitable for improving both accuracy and diversity, with the three modules.
3. The method has achieved better diversity compared with baselines.

The weaknesses of this paper are as follows.
1. The authors should compare the proposed method with more diverse recommendation algorithms.
2. The authors should use more metrics for recommendation diversity, such as entropy.
3. It is a bit strange to call a method an "accuracy-driven" model. Actually, all recommendation methods target better recommendation accuracy.

**Questions:**

Please answer my three questions above.

**Reviewer Confidence:**

4: The reviewer is certain that the evaluation is correct and very familiar with the relevant literature

**Scope:**

4: The work is relevant to the Web and to the track, and is of broad interest to the community

---

### Official Review · Reviewer_ZB42 · 2023-11-29

**Novelty:** 6
**Technical Quality:** 6

**Review:**

The paper introduces a video recommendation framework called Category based and Popularity guided Video Game Recommender System (CPGRec), which addresses critical challenges in video game recommendations. Concretely, the approach aims to improve accuracy through cross-category connections as well as diversity of game recommendations by balancing recommendations of both popular and niche games. The methodology is based on proposing a game connection module for improving accuracy by establishing relationships between game, a neighbor aggregation and nodes reweighting strategy for enhancing diversity, and a combined training model that integrates all components while focusing on negative-sample score reweighting. Experiments are conducted on a Steam dataset and illustrate how the presented work can outperform relevant baselines.

Pro:
- The paper tackles a relevant problem and is a good fit for the conference.
- The paper presents a novel approach to game recommendation systems by integrating a balance-oriented framework that specifically aims to address both accuracy and diversity in recommendations.
- The paper includes a sensitivity analysis, which shows how different parameters affect the performance of the recommendation system.
-  The empirical results show that the CPGRec outperforms several competitive baseline methods.
- The paper links to a GitHub repository that holds the dataset and the source code used to produce the experiments
- The introduction of negative-sample score reweighting is innovative to distinguish less popular games.
- While the framework itself is tested in a games recommendation scenario, its architecture and conceptual ideas could potentially be applied to other types of recommendation systems where the aim is to balance accuracy and diversity of recommendations.

Suggestions for improvement:
- The evaluation is performed on a single dataset from Steam. It is questionable how well the framework generalizes to other datasets, where gaming behavior and gaming preferences could be very different. Performing experiments across diverse datasets would help to better understand the robustness of the system against different types of gaming preferences and behaviors.
- There could be a risk of overemphasizing long-tail games at the expense of what users might prefer or expect to see, potentially leading to a disconnect between the system's suggestions and user satisfaction - this is not really discussed in the paper.

**Questions:**

What is the author's experience or intuition of how well the framework generalizes to other datasets or even recommendation scenarios beyond gaming?
What is the influence of such an approach on user satisfaction, do the authors have some insights here?

**Ethics Review Description:**

No ethical issues identified.

**Reviewer Confidence:**

3: The reviewer is confident but not certain that the evaluation is correct

**Scope:**

4: The work is relevant to the Web and to the track, and is of broad interest to the community

---

### Official Review · Reviewer_EXQi · 2023-11-30

**Novelty:** 4
**Technical Quality:** 5

**Review:**

The paper introduces CPGRec, a novel framework comprising three key components: accuracy-driven, diversity-driven, and comprehensive modules. The accuracy-driven module extends state-of-the-art game recommendation methods by creating more stringent connections among games to enhance recommendation accuracy. The diversity-driven module focuses on connecting neighbors with diverse categories within the proposed game graph and harnesses the advantages of popular game nodes to amplify the influence of long-tail games within the player-game bipartite graph, thereby enriching recommendation diversity. The comprehensive module combines these approaches and employs a new negative-sample rating score reweighting method to balance accuracy and diversity. Authors propose novel methods on different modules to promote both accuracy and diversity in game recommendation.

Advantages:
1. The paper is clearly written and easy to follow
2. They have new insights into every aspect of the model, including the transformation from raw to strict connections and the diverse-category neighbors.
3. The experiments are sufficient to show the advantages of their model.

Disadvantages:
1. Missing closely related paper
2. The novelty of the paper is slightly changes within already existing methods

**Questions:**

1. The study problem and solution of this paper are very similar to [1]. It is strange authors do not mention the paper.
2. In the model design, authors make the popular game nodes as “vehicles for propagating messages of long-tail games”. This method is quite strange for me, and the long-tail games are not necessarily connected to the popular game nodes.


[1] Liu, Kangzhe, Jianghong Ma, Shanshan Feng, Haijun Zhang, and Zhao Zhang. "DRGame: Diversified Recommendation for Multi-category Video Games with Balanced Implicit Preferences." arXiv preprint arXiv:2308.15823 (2023).

**Reviewer Confidence:**

3: The reviewer is confident but not certain that the evaluation is correct

**Scope:**

4: The work is relevant to the Web and to the track, and is of broad interest to the community

---

### Decision · Program_Chairs · 2024-01-22

**Decision:**

Accept (Oral)

**Comment:**

Quality:
 + Clear writing.
 + Novel approach.
 + Important problem is addressed.
 + Authors provide convincing responses to reviewers' questions and concerns.
 - Only a single dataset from Steam is used in evaluation.
 - Unconventional coverage and diversity metrics are used.

 Clarity:
 + Paper is clearly written and easy to follow.
 + Authors spent a considerable amount of time to provide convincing responses to reviewers' questions and concerns. They conducted new experiments and reported corresponding results, which substantially helped mitigate the concerns.

 Originality:
 + Approach seems novel.

 Significance:
 + The work is highly relevant to the recommender systems community, and therefore also to WEB.